# Trends and Patterns of Daily Maximum, Minimum and Mean Temperature in Brazil from 2000 to 2020

**Leone Francisco Amorim Curado** [1], **Sérgio Roberto de Paulo** [1], **Iramaia Jorge Cabral de Paulo** [1],
**Daniela de Oliveira Maionchi** [1], **Haline Josefa Araujo da Silva** [1], **Rayanna de Oliveira Costa** [1],
**Ian Maxime Cordeiro Barros da Silva** [1], **João Basso Marques** [1], **André Matheus de Souza Lima** [1]
**and Thiago Rangel Rodrigues** [2,*]

[1] Graduate Program in Environmental Physics, Institute of Physics, Federal University of Mato Grosso (UFMT), Cuiabá 78060-900, MT, Brazil; leone.curado@fisica.ufmt.br (L.F.A.C.); sergio@fisica.ufmt.br (S.R.d.P.); ira@fisica.ufmt.br (I.J.C.d.P.); dmaionchi@fisica.ufmt.br (D.d.O.M.); haline.jas@fisica.ufmt.br (H.J.A.d.S.); rayanna_costa@fisica.ufmt.br (R.d.O.C.); iansilva@fisica.ufmt.br (I.M.C.B.d.S.); joao.marques@ufmt.br (J.B.M.); andre.souza@fisica.ufmt.br (A.M.d.S.L.)

[2] Laboratory of Atmospheric Sciences (LCA), Federal University of Mato Grosso do Sul (UFMS), Campo Grande 79070-900, MS, Brazil

* Correspondence: thiago.r.rodrigues@ufms.br

**Abstract:** According to data obtained from meteorological towers, Brazil has significantly increased temperature in the past 20 years, particularly in the North and Midwest regions. Vapor pressure deficit and evapotranspiration were also analyzed, showing an increase across the entire country, confirming that the air is becoming drier. This warming trend is part of the global climate change phenomenon caused by the rise of greenhouse gases in the atmosphere, fires, poor soil management practices, deforestation, and logging. The increase in temperature and dryness has profoundly impacted Brazil's climate and ecosystems, leading to intensified extreme weather events and changes in the distribution of both animal and plant species. This study highlights the importance of utilizing meteorological tower data to monitor and understand the effects of climate change in Brazil. It emphasizes the need for immediate action to address its causes and mitigate its negative impacts.

**Keywords:** climate change; meteorological towers; temperature increase; dryness increase





## 1. Introduction

Studies of changes in the global climate, as well as climate variability, have been a matter of great interest to the scientific community due to the adverse impacts caused on biodiversity, water resources, energy generation, agricultural production, and even irreversible damage to society [1–3]. One of the dominant factors affecting weather patterns around the world is the El Niño Southern Oscillation (ENSO). This climatic phenomenon is known for oceanic and atmospheric variations characterized by the abnormal increase in the waters of the equatorial Pacific Ocean (El Niño) and by the abnormal cooling of the waters of the tropical Pacific Ocean (La Ninã). The ENSO is an index related to droughts, floods variations of temperature, and precipitation around the world [3,4].

Future climate projections point to a global increase in droughts' occurrence, intensity, and duration [5]. In Brazil, droughts in the northeast region tend to intensify due to reductions in precipitation due to climate change. Similarly, areas such as the Southeast and Midwest may experience substantial reductions in rainfall due to the decrease in evapotranspiration caused by deforestation in Amazon [6,7]. A drought is a climate event of an insidious nature because it develops slowly. However, it is not easy to have a universal definition. Drought gradually increases in intensity and duration with dangerous consequences for humankind and the environment. Around the world, many places

are experiencing droughts due to undeniable variations in climate, causing various eco-hydrological and socio-economic impacts, including increased risk of wildfires, water shortages, loss of crops and livestock, migration, and indirect health effects [8,9].

The application of drought indices has been a major concern for scholars and scientists [9]. No global drought index could provide universally accepted results since almost all of these indices are based on observed data. It is crucial to have a better sense of how drought can develop and how it can be monitored [10]. Drought can be classified into four categories: (i) meteorological (such as rainfall deficit); (ii) agricultural (such as lack of moisture in the soil); (iii) hydrological (tracked considering the decrease in flows and runoff); and finally (iv) socioeconomic droughts on the human use of water, while there is also another definition of drought based on ecological water deficit in the environment as ecological drought [8,10].

The main factor influencing extreme events related to climate change, such as droughts, floods, and storms, can be explained by the increase in global warming that can affect the hydrological cycle [8,9]. The fact that drought is strongly linked to climatological events means that parameters such as precipitation and air temperature can serve as reliable indicators of drought occurrence. Therefore, transforming drought indicators into indices is essential [9].

It is evident that warming induced by climate change has accelerated hydrological processes, firstly by increasing the energy available for evapotranspiration (ET) and secondly by rising temperatures and therefore the water holding capacity of the atmosphere [8]. Saturation vapor pressure, the amount of water vapor in the air, is a nonlinear function of air temperature. Thus, increasing global Earth surface temperature is increasing the saturation vapor pressure of the atmosphere [11]. The vapor pressure deficit (VPD) is a determining variable of global water resources and plant water relations. Due to its coupling with temperature, it may become increasingly crucial for vegetation dynamics in the coming decades [11]. It is expected that the VPD will increase in the continents in a few decades due to its relationship with the increase in temperature and, depending on the region, with the decrease in relative humidity [12]).

ET is a component that is still not fully understood. Its importance is demonstrated by its participation in the hydrological cycle and the surface energy balance. Given that a high percentage of precipitated water is evaporated or transpired, water budgets are dictated by ET fluctuations and subsequently by ET's dependence on various environmental parameters [13].

Vegetation in different climate zones and terrestrial biomes may show relatively different responses to drought at different time scales [14]. When occurring in humid, semi-arid, or arid climates, evapotranspiration will remain the same, decrease, or cease once it has faced drought conditions in the past. Therefore, how an ecosystem responds to dry weather depends on its previous adaptation and the extent of environmental forcing [15].

Thus, the general objective of this article is to analyze the patterns and trends of drought rates in Brazilian tropical ecosystems in the last 20 years. For this, the annual behavior of air temperatures (maximum, average, and minimum), the vapor pressure deficit, reference evapotranspiration, and precipitation for all regions of Brazil are analyzed.

## 2. Material and Methods

The study was conducted in all Brazilian states (Figure 1). The data were divided according to the political division of Brazil in five regions, namely North, Northeast, Midwest, Southeast, and South. Temperature data (maximum, average, and minimum)precipitation and relative humidity were obtained through the meteorological towers of the National Institute of Meteorology—INMET. (https://portal.inmet.gov.br/, accessed on 7 February 2023). The study period of this research was from 2000 to 2020, and the towers used are described in Annex 1, with a total of 152 towers.

The vapor pressure deficit (VPD) in kPa, was calculated from the following equation:

$$VPD = e_s - e, \tag{1}$$

where $e_s$ is the saturation pressure, and $e$ is the current vapor pressure in the atmosphere, given by

$$e_s = 0.61078 \times 10^{7.5T/(237.3+T)}, \qquad e = e_s \frac{RH}{100}, \qquad (2)$$

where $T$ is the air temperature (°C) and $RH$ is the relative humidity (%), measured in the meteorological tower. The reference evapotranspiration ($ET_0$), in mm/day, was calculated using the Garcia–Lopes equation,

$$ET_0 = \left[1.21 \times 10^{7.45T/(243.7+T)} \left(1 - \frac{RH}{100}\right) + 0.21T\right] - 2.3. \qquad (3)$$

The precipitation data were analyzed in annual terms. The months were considered dry for accumulated precipitation below 50 mm and with heavy rainfall, when precipitation was more significant than 300 mm in the North region and 200 mm in the South region. These criteria were chosen based on observations of the maximum and minimum values of monthly accumulated precipitation in the regions studied.

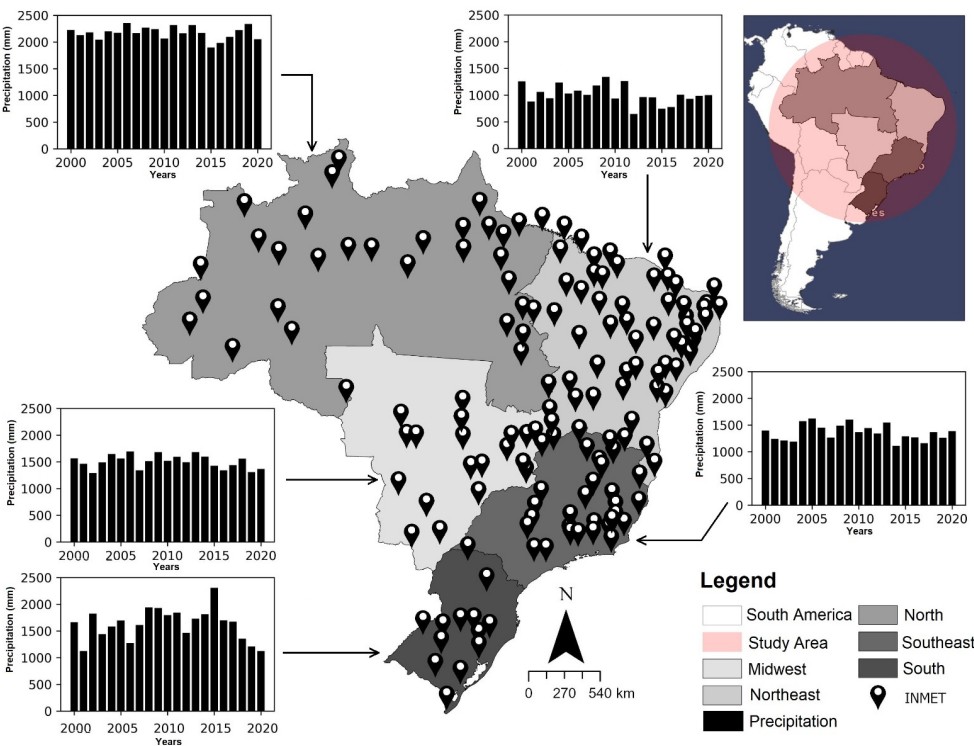

**Figure 1.** Map of Brazil with the five regions, annual accumulated precipitation, and locations of the meteorological towers used.

In this study, the trends of all variables were analyzed using a linear framework, specifically by considering a first-order relationship between the variables and time. Throughout the analysis, the coefficient of determination ($R^2$) was computed to assess the significance of the linear model. The temperature change rates were determined through the angular coefficient of the linear fit, along with its corresponding confidence interval.

## 3. Results

### 3.1. Regional Analysis

Observing the annual averages of the northern region of Brazil (Table 1), there was a significant increase in temperature, in three aspects: minimum, average, and maximum, with the highest value in 2016 for average and maximum temperatures and in 2015 for minimum temperature. The minimum temperature increased the most over the 20 years of

the study. It was also observed that the minimum and average temperatures of this region recorded the lowest value at the beginning of the study series (the year 2000), indicating that the region has been experiencing a trend of increasing temperatures throughout the period considered. The temperature augmentation was approximately 1 °C, exceeding the expectations of the Intergovernmental Panel on Climate Change, which presented an increase between 0.5 °C and 1 °C between 2015 and 2039 for South American temperatures (IPCC, 2014). In recent works [16,17], also reported an increase of 4 °C in temperature in the region under consideration.

**Table 1.** Average annual temperature values (maximum, average, and minimum), vapor pressure deficit, reference evapotranspiration during 20 years and an average of the decades (2000 to 2009 and 2010 to 2020) in the Brazilian North, Northeast, and Midwest regions. The highest values of the series are shown in red and the lowest in bold.

| Year | North Region | | | | | Northeast Region | | | | | Midwest region | | | | |
|---|---|---|---|---|---|---|---|---|---|---|---|---|---|---|---|
| | Tmax | Tavg | Tmin | VPD | ET0 | Tmax | Tavg | Tmin | VPD | ET0 | Tmax | Tavg | Tmin | VPD | ET0 |
| 2000 | **32.20** | **26.30** | **22.22** | **0.62** | **4.38** | **30.99** | **25.36** | **21.07** | **0.85** | **4.62** | **30.20** | **23.50** | 18.53 | **0.85** | **4.24** |
| 2001 | 32.52 | 26.54 | 22.41 | 0.63 | 4.46 | 31.75 | 25.82 | 21.23 | 1.01 | 5.02 | 30.50 | 23.69 | 18.62 | 0.88 | 4.33 |
| 2002 | 32.78 | 26.82 | 22.64 | 0.69 | 4.63 | 31.68 | 25.86 | 21.41 | 1.01 | 5.02 | 31.16 | 24.28 | 19.05 | 1.00 | 4.69 |
| 2003 | 32.75 | 26.88 | 22.77 | 0.70 | 4.66 | 31.89 | 26.09 | 21.58 | 1.05 | 5.15 | 30.40 | 23.62 | 18.59 | 0.90 | 4.35 |
| 2004 | 32.59 | 26.75 | 22.67 | 0.69 | 4.62 | 31.42 | 25.79 | 21.37 | 0.96 | 4.91 | 30.28 | 23.60 | 18.68 | 0.89 | 4.33 |
| 2005 | 33.19 | 27.13 | 22.92 | 0.77 | 4.85 | 31.77 | 26.10 | 21.64 | 1.02 | 5.10 | 30.59 | 23.80 | 18.85 | 0.92 | 4.43 |
| 2006 | 32.67 | 26.80 | 22.71 | 0.71 | 4.66 | 31.61 | 25.94 | 21.48 | 0.99 | 5.01 | 30.50 | 23.80 | 18.89 | 0.86 | 4.31 |
| 2007 | 32.87 | 26.88 | 22.60 | 0.75 | 4.76 | 31.68 | 25.93 | 21.27 | 1.06 | 5.13 | 31.38 | 24.24 | 18.78 | 1.05 | 4.77 |
| 2008 | 32.53 | 26.64 | 22.54 | 0.70 | 4.61 | 31.58 | 25.87 | 21.24 | 1.02 | 5.05 | 30.66 | 23.65 | **18.42** | 0.96 | 4.47 |
| 2009 | 32.85 | 27.11 | 23.06 | 0.71 | 4.73 | 31.62 | 26.01 | 21.62 | 0.94 | 4.93 | 30.79 | 23.96 | 19.04 | 0.86 | 4.35 |
| 2010 | 33.27 | 27.28 | 23.07 | 0.75 | 4.84 | 32.27 | 26.49 | 21.88 | 1.06 | 5.25 | 31.34 | 24.04 | 18.59 | 1.02 | 4.67 |
| 2011 | 32.66 | 26.77 | 22.66 | 0.69 | 4.61 | 31.48 | 25.80 | 21.29 | 0.94 | 4.88 | 30.62 | 23.66 | 18.46 | 0.93 | 4.42 |
| 2012 | 32.64 | 26.79 | 22.67 | 0.70 | 4.64 | 32.41 | 26.42 | 21.66 | <span style="color:red">1.17</span> | 5.45 | 30.97 | 23.97 | 18.78 | 0.97 | 4.56 |
| 2013 | 32.66 | 26.94 | 22.99 | 0.68 | 4.64 | 32.17 | 26.46 | 21.89 | 1.10 | 5.31 | 30.72 | 23.90 | 18.87 | 0.89 | 4.40 |
| 2014 | 32.50 | 26.86 | 22.96 | 0.67 | 4.59 | 31.79 | 26.09 | 21.41 | 1.01 | 5.08 | 31.22 | 24.19 | 19.01 | 0.95 | 4.58 |
| 2015 | 33.17 | 27.35 | <span style="color:red">23.24</span> | 0.77 | 4.89 | 32.66 | 26.67 | 21.76 | 1.17 | 5.49 | <span style="color:red">31.86</span> | <span style="color:red">24.62</span> | 19.19 | 1.00 | 4.76 |
| 2016 | <span style="color:red">33.40</span> | <span style="color:red">27.43</span> | 23.06 | <span style="color:red">0.78</span> | <span style="color:red">4.92</span> | <span style="color:red">32.76</span> | <span style="color:red">26.86</span> | <span style="color:red">21.98</span> | 1.15 | <span style="color:red">5.50</span> | 31.68 | 24.43 | 19.13 | 1.04 | 4.79 |
| 2017 | 32.95 | 27.11 | 22.99 | 0.73 | 4.76 | 32.07 | 26.38 | 21.69 | 1.06 | 5.23 | 31.59 | 24.48 | 19.30 | 1.00 | 4.72 |
| 2018 | 32.90 | 27.04 | 22.94 | 0.74 | 4.76 | 32.22 | 26.35 | 21.55 | 1.04 | 5.19 | 31.13 | 24.28 | 19.42 | 0.91 | 4.52 |
| 2019 | 32.95 | 27.20 | 23.10 | 0.70 | 4.73 | 32.45 | 26.53 | 21.65 | 1.03 | 5.20 | 31.70 | 24.62 | <span style="color:red">19.45</span> | 1.06 | 4.87 |
| 2020 | 33.27 | 27.37 | 23.10 | 0.77 | 4.89 | 32.05 | 26.42 | 21.74 | 0.87 | 4.82 | 31.45 | 24.63 | 19.44 | <span style="color:red">1.11</span> | <span style="color:red">4.96</span> |
| 2000–2009 | 32.69 | 26.78 | 22.65 | 0.70 | 4.60 | 31.60 | 25.88 | 21.39 | 0.99 | 4.99 | 30.65 | 23.81 | 18.74 | 0.92 | 4.43 |
| 2010–2020 | 32.94 | 27.11 | 22.98 | 0.73 | 4.75 | 32.21 | 26.41 | 21.68 | 1.05 | 5.22 | 31.30 | 24.26 | 19.06 | 0.99 | 4.66 |

The VPD and $ET_0$ variables also rose during the analysis period, with higher values in 2016 and lower values in 2000, indicating that there is a gradual increase in these variables over the years in the Amazon region. These trends corroborate with the work of [16] who found an increase in evapotranspiration in this region, while [11], reported an increase in global VPD in recent decades and an upward trend towards the future. According to land use and occupation maps, there was also an increase in deforested areas in the region during the period of study, which agrees with [17].

Another important analysis is the annual mean values between the two decades, where all the variables showed an increase in the second decade, except in the South region, see Tables 1 and 2. We emphasize that the Northeast and Southeast regions presented an increase of 0.6 °C in the maximum temperature and the Northeast and Midwest regions presented an increase of 0.53 °C and 0.45 °C, respectively, in the average temperature. All the highest values of the analyzed data occurred in the second decade, in the four regions, and the lowest values in the first decade, with the exception of the minimum temperature and the VPD in the Southeast region.

**Table 2.** Average annual temperature values (maximum, average, and minimum), vapor pressure deficit, reference evapotranspiration during 20 years, and the average of the decades (2000 to 2009 and 2010 to 2020) in the Brazilian Southeast and South regions. The highest values of the series are shown in red and the lowest in bold.

| Year | Southeast Region | | | | | South Region | | | | |
|---|---|---|---|---|---|---|---|---|---|---|
| | Tmax | Tavg | Tmin | VPD | ET0 | Tmax | Tavg | Tmin | VPD | ET0 |
| 2000 | 28.69 | 22.53 | 18.00 | 0.76 | **3.77** | 24.02 | **17.94** | **13.55** | 0.57 | 2.54 |
| 2001 | 29.40 | 23.06 | 18.37 | 0.74 | 3.96 | 25.44 | 19.16 | 14.90 | 0.53 | 2.72 |
| 2002 | 29.62 | 23.23 | 18.70 | 0.74 | 3.98 | 24.99 | 19.11 | 14.85 | 0.55 | 2.76 |
| 2003 | 29.35 | 22.90 | 18.25 | 0.82 | 3.97 | 24.92 | 18.67 | 14.12 | 0.57 | 2.71 |
| 2004 | **28.28** | **22.29** | 17.95 | 0.67 | 3.65 | 24.69 | 18.42 | 13.76 | 0.59 | 2.69 |
| 2005 | 28.87 | 22.88 | 18.55 | 0.70 | 3.83 | 25.02 | 18.92 | 14.46 | 0.62 | 2.85 |
| 2006 | 28.93 | 22.63 | 18.05 | 0.72 | 3.78 | 25.27 | 18.89 | 14.24 | 0.62 | 2.83 |
| 2007 | 29.52 | 23.04 | 18.25 | 0.86 | 4.03 | 25.13 | 18.91 | 14.38 | 0.59 | 2.79 |
| 2008 | 28.85 | 22.49 | 17.93 | 0.79 | 3.79 | 24.42 | 18.25 | 13.75 | 0.55 | 2.57 |
| 2009 | 29.20 | 23.08 | 18.64 | 0.71 | 3.96 | 24.66 | 18.55 | 14.15 | 0.51 | 2.56 |
| 2010 | 29.43 | 22.95 | 18.13 | 0.80 | 3.96 | **23.64** | 18.10 | 14.08 | 0.48 | **2.42** |
| 2011 | 28.81 | 22.47 | **17.80** | 0.79 | 3.76 | 23.66 | 18.00 | 13.98 | 0.50 | 2.43 |
| 2012 | 29.43 | 23.02 | 18.32 | 0.78 | 4.03 | 24.78 | 18.90 | 14.67 | 0.57 | 2.75 |
| 2013 | 29.03 | 22.78 | 18.45 | 0.71 | 3.85 | 23.70 | 18.03 | 13.90 | 0.49 | 2.42 |
| 2014 | 29.79 | 23.27 | 18.46 | 0.88 | 4.19 | 24.72 | 19.14 | 15.18 | 0.51 | 2.69 |
| 2015 | 30.19 | 23.72 | 19.03 | 0.91 | 4.26 | 24.35 | 18.94 | 15.21 | **0.46** | 2.55 |
| 2016 | 30.11 | 23.54 | 18.82 | 0.89 | 4.21 | 23.68 | 18.08 | 14.09 | 0.50 | 2.45 |
| 2017 | 29.55 | 23.09 | 18.34 | 0.84 | 4.09 | 24.86 | 19.23 | 15.24 | 0.54 | 2.76 |
| 2018 | 29.51 | 23.12 | 18.47 | 0.75 | 4.08 | 24.58 | 18.95 | 14.89 | 0.52 | 2.66 |
| 2019 | 30.41 | 23.83 | 18.89 | 0.78 | 4.26 | 24.73 | 19.12 | 15.09 | 0.55 | 2.76 |
| 2020 | 30.10 | 23.81 | 19.17 | **0.65** | 4.18 | 24.73 | 18.73 | 14.25 | 0.61 | 2.80 |
| 2000–2009 | 29.07 | 22.81 | 18.27 | 0.75 | 3.87 | 24.85 | 18.68 | 14.22 | 0.569 | 2.70 |
| 2010–2020 | 29.67 | 23.24 | 18.53 | 0.80 | 4.08 | 24.31 | 18.66 | 14.60 | 0.521 | 2.61 |

As for precipitation, there was no change in the accumulated regional average between the years studied, as can be seen in Figure 1. However, in states such as Amazonas, there was an increase in the number of months with more than 300 mm of rain, going from 23 months in the first decade to 32 months in the second one. It was also noticed that in the first decade, 80% of the years had 2 or fewer months above 300 mm, while in the second decade, about 70% of the years had more than 2 months with rain above 300 mm, especially in 2011 and 2019, which recorded 5 months each with heavy rains. These results corroborate with [16].

The Northeast region also showed an increment in air temperatures and $ET_0$, with a greater increase in average and maximum temperature between 2000 and 2020 of 0.5 °C (mean square deviation, $R^2 = 0.61$) and 1°C ($R^2 = 0.50$), respectively. The highest values of the time series also occurred in the second decade of the study, and the lowest values at the beginning, indicating that this region also undergoes a progressive increase in air temperature and $ET_0$, which also corroborate with [18].

Analyzing the precipitation in the Northeast region, a pattern of reduction in its annual accumulation was observed in the period considered. Besides that, in the first decade, there are eight years with accumulated rainfall above 1000 mm annually, while in the second decade only 1 year (2011) with rains above this value. Another factor is the number of months considered dry, with accumulated precipitation below 50 mm. It increased from 48 months in the first decade to approximately 58 in the second decade, indicating that the region is experiencing an increase in dry months. Emphasis is given to the state of Pernambuco, which presented an increase of 21 months of drought in the second decade. Ref. [18] also reports that there was a decrease in rainfall in Northeast Brazil.

*3.2. Local Analysis*

3.2.1. General Aspects

The places that presented the greatest variations in each region in the two decades considered are described below.

In the North region, the greatest variation in maximum temperature of 1.90 °C ($R^2 = 0.66$) occurred in Palmas, Tocantins, a state that is in the transition between the Amazon Forest, Caatinga, and Cerrado, with the highest mean value in 2016 reaching 35.67 °C, 2.93 °C higher than in the beginning of the series (32.74 °C). The average temperature also had a greater increase in the Tocantins, at the Araguainha station, of 1.33 °C between the years 2000 and 2020 ($R^2 = 0.68$), with the highest annual mean value also in 2016 (26.22 °C). The station with the greatest variation in the minimum temperature was Caracarai, state of Roraima, with an increase of 1.81°C ($R^2 = 0.74$); with the highest annual average in 2016 (24.51 °C) and the lowest average value in 2004 (22.10 °C); difference of 2.41 °C between the extremes of the series. This region is marked by the transition of the Amazon Forest and agricultural crops. It should also be noted that the greatest variation in VPD ($R^2 = 0.75$) and $ET_0$ ($R^2 = 0.77$) in the North region occurred at the same station, Porto de Moz in the state of Pará, with a progressive increase between 2000 and 2020, and higher values in 2015 and lower values in 2000.

In the Northeast region, the stations that presented the greatest variations during the study period were Balsas in Maranhão ($R^2 = 0.63$) for maximum temperature, Petrolina in Pernambuco ($R^2 = 0.52$) for average temperature and Natal in Rio Grande do Norte ($R^2 = 0.44$) for minimum temperature, the difference between the values at the beginning and end of the data series were 1.98 °C, 1.77 °C, and 4.40 °C, respectively. The years with the highest average values were 2016 for maximum and minimum temperatures and 2019 for average temperature, and the year with the lowest average value was 2000 (beginning of the series) for all temperature aspects, comparing the annual average values of maximums and minimum, there was an increase of 2.94 °C, 1.77 °C, and 5.28 °C in the values of maximum, average, and minimum temperature, respectively. The "Própria" station in Sergipe, on the other hand, presented the highest $ET_0$ variation ($R^2 = 0.55$), with an increase of approximately 1 mm·day$^{-1}$ between 2000 and 2020.

In the Midwest region, the stations that showed the greatest increases were: Jataí for maximum temperature ($R^2 = 0.69$) and average ($R^2 = 0.63$) and Formosa for minimum temperature ($R^2 = 0.61$). The variations between 2000 and 2020 were 1.35 °C for maximum, 0.93 °C for average, and 1.23 °C for the minimum. Comparing the highest and lowest annual values all occurred in 2019 and 2001, respectively, the increase was 1.39 °C, 1.21 °C and 2.01 °C, respectively. The Goiânia station showed the greatest increase in VPD ($R^2 = 0.54$) and $ET_0$ ($R^2 = 0.56$). An important factor to highlight is that all the stations with

the highest increases in this region belong to the state of Goiás, which has undergone an intense transformation of the soil through crops and cattle raising.

In the Southeast region, the Boa Esperança station, in the state of Espírito Santo, presented greater annual variations in maximum ($R^2 = 0.46$) and minimum ($R^2 = 0.29$) temperatures, with differences between 2000 and 2020 of 2.49 °C and 1.75 °C, respectively, with the highest value in 2016 for maximum temperature (33.18 °C) and 2020 for minimum temperature (20.64 °C). The average temperature showed the highest annual variation ($R^2 = 0.49$) at the Sorocaba station, São Paulo, with an annual average difference of 1.88 °C between 2000 and 2020. The Salinas station, in Minas Gerais, showed the highest increase in $ET_0$ ($R^2 = 0.40$), with the highest average annual difference recorded between 2016 (highest) and 2004 (lowest) of 1.60 mm/day.

### 3.2.2. Temperature

The North and Midwest regions presented the biggest variations in temperature in the period 2000–2020. The annual mean temperature raised 1 °C and the absolute maximum value of the temperature in each year 4 °C in both regions, as can be seen in Figures 2 and 3.

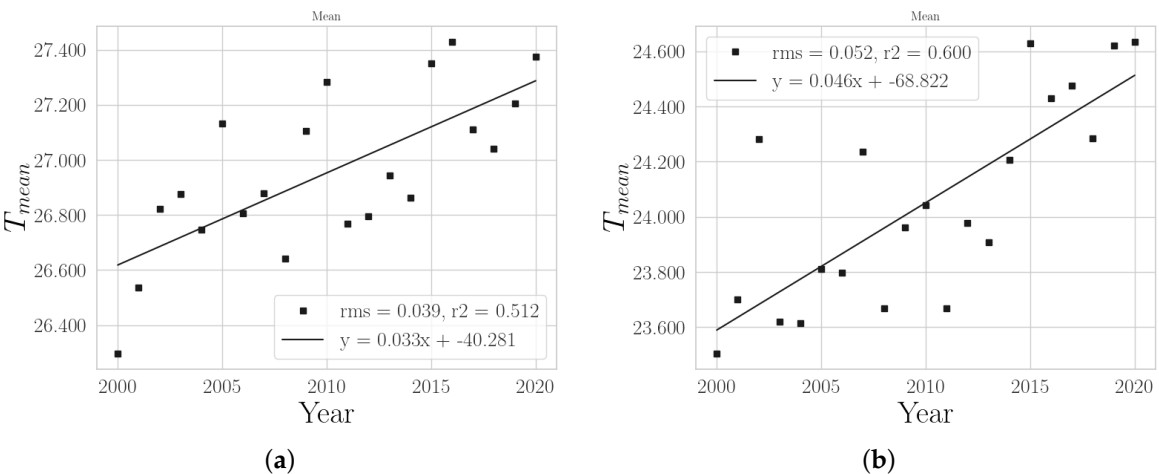

**Figure 2.** Annual mean temperature in (**a**) North and (**b**) Midwest regions.

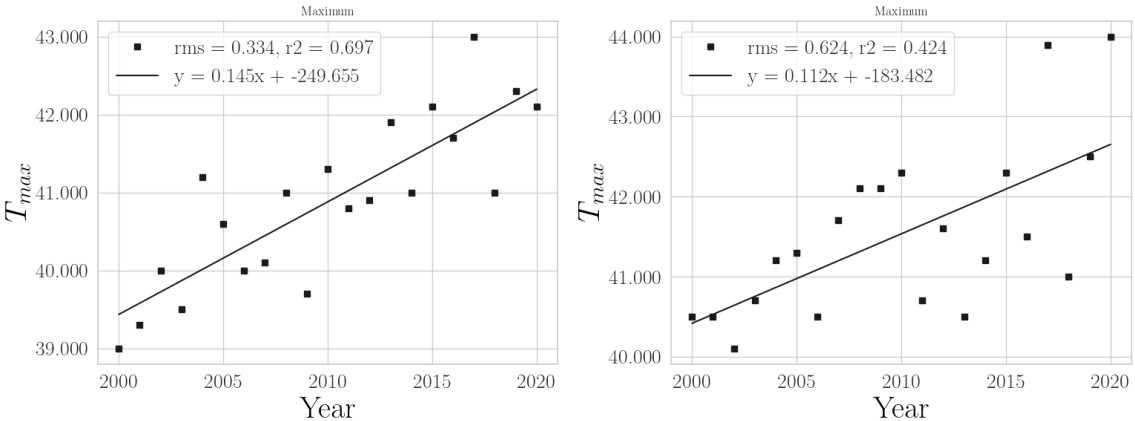

**Figure 3.** Annual highest temperature in North and Midwest regions.

In the Midwest, the minimum of $T_{max}$ increased by approximately 4 °C, while the minimum of $T_{min}$ rose by around 6 °C. Meanwhile, in the North, the minimum values of $T_{max}$ and $T_{min}$ oscillated with no definite pattern. This could be due to the higher presence of forests in the Amazon region, while in the Midwest region, the Cerrado and agricultural activities predominate. In the Northeast region, the annual mean temperature raised 1 °C, and the mean of $T_{max}$ almost 3 °C, as shown in Figure 4.

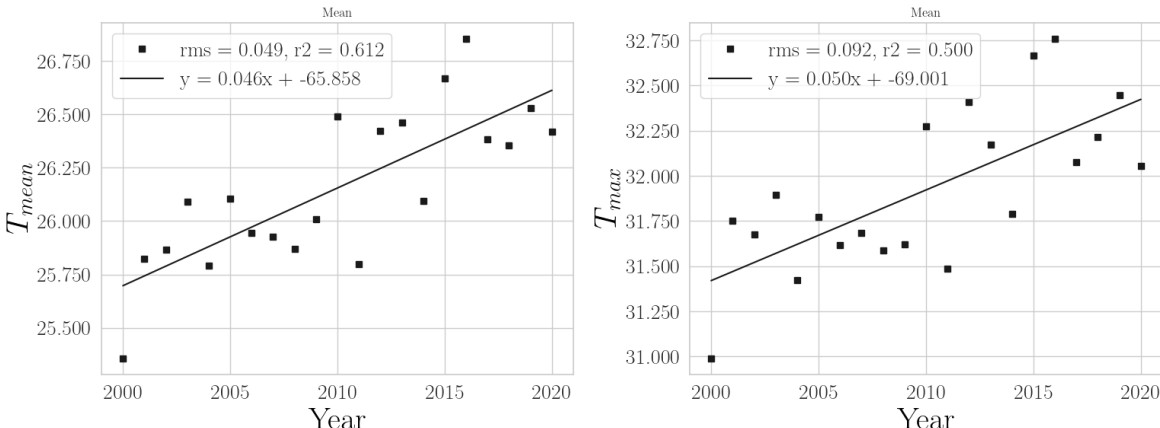

**Figure 4.** Annual mean of the daily temperature and daily maximum temperature in Northeast.

As shown in Figure 5, the annual mean temperature, as well as the mean of $T_{max}$, also increased in the Southeast around 1 °C in the last 20 years, although the highest temperatures and the lowest ones remain oscillating. In the South region, the highest temperatures are increasing while the lowest is decreasing, as can be seen in Figure 6.

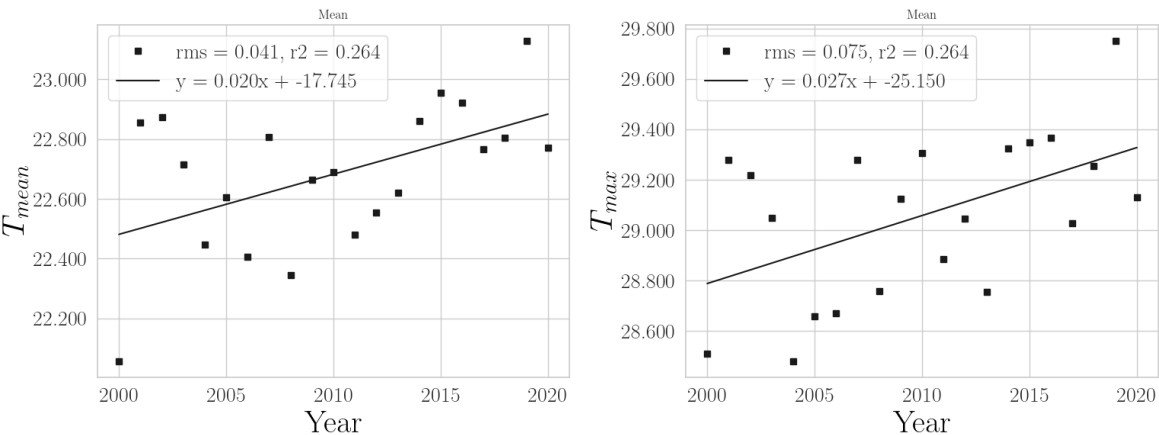

**Figure 5.** Annual mean of the daily temperature and daily maximum temperature in Southeast.

For all regions, the temperature quartiles follow the exact behavior of the mean, maximum, and minimum cases. These and all other graphics related to temperature in all regions can be consulted in Appendix A. Table 3 presents statistical analyses for linear regressions.

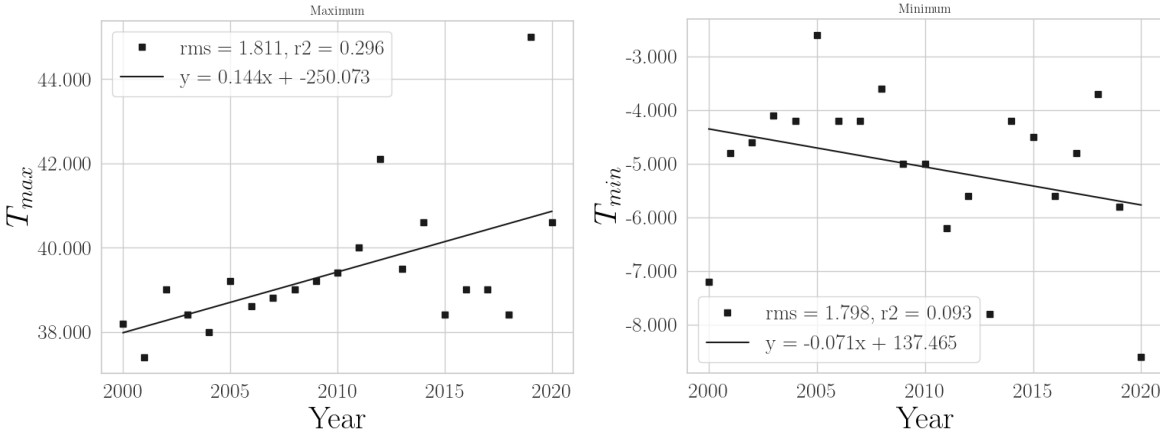

**Figure 6.** Annual highest and lowest temperature in South.

**Table 3.** Linear regression analyses for intercept and slope parameters (T-Test value) and root mean square error (RMSE).

| | | Tmax | | | | Tmean | | | | Tmin | | |
|---|---|---|---|---|---|---|---|---|---|---|---|---|
| Region | $R^2$ | Intercept | Slope | RMSE | $R^2$ | Intercept | Slope | RMSE | $R^2$ | Intercept | Slope | RMSE |
| North | 0.697 | 0 *** | 0.012 * | 5.369 | 0.512 | 0 *** | 0.0003 *** | 4.456 | 0.584 | 0 *** | 0.0001 *** | 4.106 |
| Midwest | 0.424 | 0 *** | 0.0001 *** | 4.189 | 0.599 | 0 *** | 0 *** | 4.032 | 0.544 | 0 *** | 0.0001 *** | 4.297 |
| Northeast | 0.5 | 0 *** | 0.0003 *** | 4.504 | 0.612 | 0 *** | 0 *** | 3.968 | 0.386 | 0 *** | 0.0027 ** | 4.99 |
| Southeast | 0.264 | 0 *** | 0.0015 ** | 4.848 | 0.264 | 0 *** | 0.0021 ** | 4.923 | 0.142 | 0 *** | 0.0114 * | 5.355 |
| South | 0.062 | 0 *** | 0.216 | 6.108 | 0.027 | 0 *** | 0.555 | 6.306 | 0.178 | 0 *** | 0.068 | 5.818 |

\* *p*-value < 0.05, \*\* *p*-value < 0.01, \*\*\* *p*-value < 0.001.

### 3.2.3. Vapor Pressure Deficit and Evapotranspiration

The North and Midwest regions showed the most significant changes in $ET_0$ and VPD from 2000 to 2020. The Midwest had an increase of 0.7 mm/day in annual mean $ET_0$, 1 mm/day in minimum value, and 2 mm/day in maximum value. The North had a 0.5 mm/day increase in mean value, 1 mm/day in minimum value, and 3 mm/day increase in maximum value. VPD in the Midwest had a 0.25 kPa increase in mean and a 0.0125 kPa increase in minimum value. In the North, VPD had a 0.15 kPa increase in mean and a 1.75 kPa increase in maximum value. Figures 7 and 8 show some cases described for $ET_0$ and VPD for Midwest and North regions.

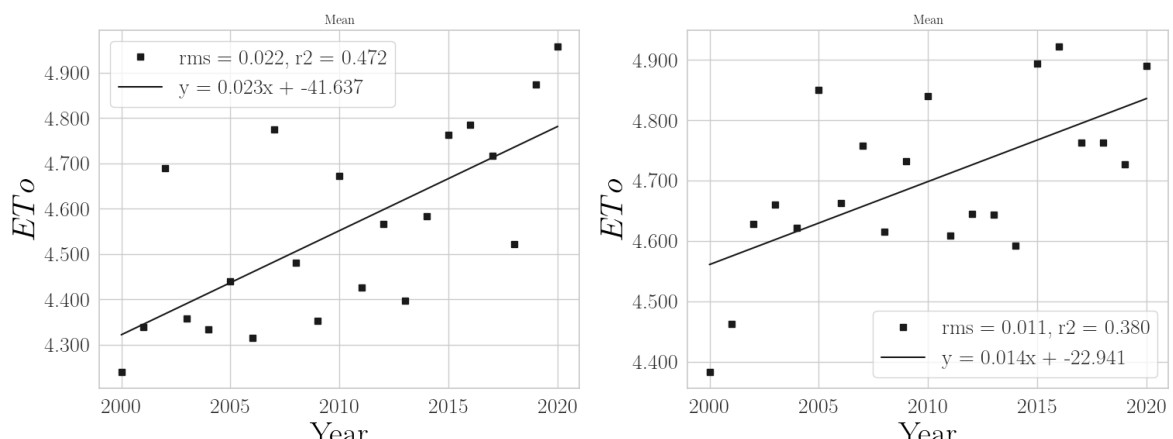

**Figure 7.** Annual mean value of $ET_o$ in Midwest and North regions.

The annual mean value of $ET_0$ and VPD in the Northeast increased by approximately 0.8 mm/day and 0.2 kPa; the maximum and minimum values show no clear pattern. The same is true for the data of $ET_0$ and VPD in Southeast and South. Further research, such as

studying the distribution of these variables throughout the months, may be necessary to understand their fluctuations better.

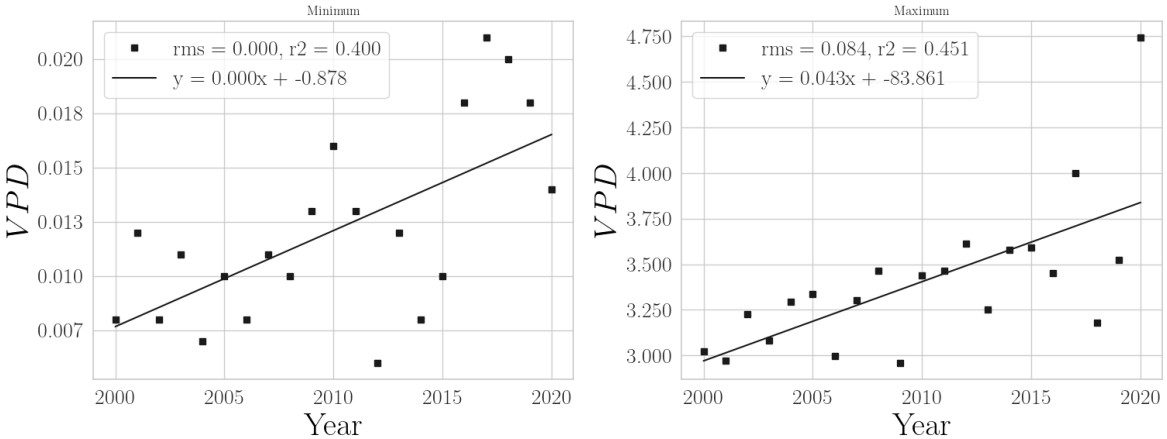

**Figure 8.** Annual lowest VDP in Midwest and highest VDP in North.

All graphics related to $ET_0$ and VPD in all regions can be consulted in Appendix A.

## 4. Discussion

The results referring to the Amazon region and the Midwest are more worrying since these regions, where forest and Cerrado vegetation predominate, were strongly affected by changes in soil use. The authors of [19] report a loss of 51.9% of forest and 49.4% of Savannah classes in these 20 years of analysis. On the other hand, the agricultural areas had the most significant increases in the same period. According to [20], the advance of agriculture and logging caused these changes in land cover. From 2008 to 2020, the Midwest region was responsible for 18.71% of deforestation in the Amazon biome and 39% in the arc of deforestation of the Brazilian Cerrado between 2011 and 2020 [21].

Ref. [22] indicated a significant upward trend in almost the entire Amazon region, which corroborates the results presented in this study. This contributed to the increase in air temperature in these regions, as reported by [19], who showed an increase in maximum and average temperatures in the Amazon region and in states in the Midwest region, such as Mato Grosso, which had the highest increase in average annual temperature.

Another factor that contributed to the changes was the fires. According to [19] and Morais et al. (2022), these regions had high rates of fire during the period studied, especially in the areas of the arc of Amazonian deforestation. Several studies [21,23–29] have pointed out changes in the use and modification of land cover in the Brazilian territory, which causes substantial impacts on temperature and other environmental variables.

Other authors [30,31], report that the change in vegetation cover is a critical factor in local climate change, affecting albedo and evapotranspiration. Thus, several factors of the water regime, especially in wetlands, such as water retention capacity and duration of flooding, can be affected by increasing temperature [32,33]. Ref. [34] also reported a loss of native vegetation cover in the Caatinga region (Northeastern Brazil) in the last 20 years. In a recent study, [35] reported an increase of up to $3°C$ in the temperature of the Amazon region. These authors also report that days and nights in this region have become warmer, possibly associated with the increase in consecutive dry days. Ref. [18] also found similar results for the Northeast region of Brazil.

The exact reason for the observed changes in temperature patterns in South Brazil cannot be determined without further information and research. Various factors could contribute to these changes, including natural climate variability, changes in atmospheric circulation patterns, human activities such as greenhouse gas emissions and deforestation, and changes in land use patterns. It is essential to conduct further research to understand these temperature changes' causes and potential impacts.

Considering Brazil as a whole, the linear regression of the complete dataset analyzed (more than 860,000 measurements for each variable) yields a rate of change in daily average temperature of $(-0.0074 \pm 0.0007)$ °C/year, indicating a slightly negative but modest intensity. However, when the data is divided into two separate subsets—1: Southern and Southeastern regions (550,000 measurements) and 2: Northern, Northeastern, and Midwest regions (310,000 measurements)—the results for the rate of change in daily average temperature are $(-0.029 \pm 0.001)$ °C/year and $(+0.046 \pm 0.007)$ °C/year, respectively. These findings are consistent with the data depicted in Appendix A graphs, which reveal a cooling trend in several analyses of Tmed (average temperature), except for the Northern, Northeastern, and Midwest regions. Therefore, it is possible to conclude that the data suggests a warming trend in the northern half of the country and a cooling trend in the southern half.

The northern half of the country is the most affected by the effects of land-use changes, particularly deforestation. The rate of warming observed for the mean daily temperature in this study can be compared to the rate of warming observed in the Mississippi region, USA, where it is also known that an intense deforestation process has been occurring in recent decades [36]. Using data provided by State Climate Summaries 2022 of NOAA National Centers (https://statesummaries.ncics.org/chapter/ms/, accessed on 24 May 2023), the rate of warming for the past 20 years is $(+0.031 \pm 0.005)$ °C/year, a value close to that found in this study for the northern half of Brazil. Considering that Mississippi is suffering a intensive deforestation process, the warning rate observed in this study in the northern half should be considered of significant importance.

It should also be considered that changes in air temperature, reference evapotranspiration, and other variables are also affected by abnormal weather events such as the El Niño Southern Oscillation—ENSO [37,38]. In the years of this phenomenon, several parts of Brazil, especially the Amazon region, experienced an increase in air temperature and evapotranspiration, more so the dry period, and a change in the rainfall regime, as reported by [3]. This contributes to increasing local and regional climate changes intensified by human action.

## 5. Conclusions

This study found a significant increase in temperature in Brazil, especially in the North and Midwest regions, which had an increase of 4 °C in the maximum temperature over the last 20 years. For all of the analysis of daily mean, maximum, and minimum temperatures depicted in Appendix A, almost all linear approaches indicate a positive rate, except some of the South and Southeast regions. Aside from that, evapotranspiration and VPD also presented an increase in their values, as is shown in the respective figures of Appendix A. $ET_0$ increased the minimum value of 1 mm/day and the maximum value of even 2 and 3 mm/day in North and Midwest regions. These variations indicate that these regions are becoming drier over the years.

The overall results of this article indicate warming in regions where deforestation occurs, highlighting the sensitivity of ecosystems to this type of land-use change. It is likely that deforestation has a direct impact on the loss of water retention capacity by ecosystems, leading to a reduced ability to absorb energy through processes associated with latent heat flux. This, in turn, results in significant changes in energy partitioning in the atmosphere, which is related to the behavior of the time series of VPD (vapor pressure deficit) and $ET_0$ (evapotranspiration).

In conclusion, this warming trend, along with rising vapor pressure deficit and evapotranspiration, indicates growing aridity across the country, linked to global climate change, which presents itself as an association of natural (El Niño) and anthropic (change in land use and cover) factors. The study emphasizes the importance of using meteorological tower data to monitor and understand these effects and the need for prompt action to tackle the causes of climate change and mitigate its harmful effects. It is imperative to curb climate change's causes and reduce its negative impact on the planet.

**Author Contributions:** Data curation, I.M.C.B.d.S., H.J.A.d.S., A.M.d.S.L., R.d.O.C.; methodology and formal analysis, T.R.R., L.F.A.C. and D.d.O.M.; writing visualization and discussion, S.R.d.P., I.J.C.d.P., L.F.A.C., J.B.M. and D.d.O.M. All authors have read and agreed to the published version of the manuscript.

**Funding:** This research received no external funding.

**Data Availability Statement:** The data that support the findings of this study are available from the corresponding author upon reasonable request.

**Acknowledgments:** The authors would like to express their gratitude to Brazilian Coordenação de Aperfeiçoamento de Pessoal de Nível Superior (CAPES), Conselho Nacional de Desenvolvimento Científico e Tecnológico (CNPq) and Pró-Reitoria de Pesquisa e Pós-Graduação da Universidade Federal de Mato Grosso do Sul for supporting this study, as well as to thank National Institute of Research in Pantanal (INPP) and the Large-Scale Biosphere-Atmosphere Program (LBA), coordinated by the National Institute for Amazonian Research (INPA), for the use and data availability, logistical support, and infrastructure during field activities.

**Conflicts of Interest:** The authors declare no conflict of interest.

## Appendix A

*Appendix A.1. Midwest*

Appendix A.1.1. Annual Temperatures

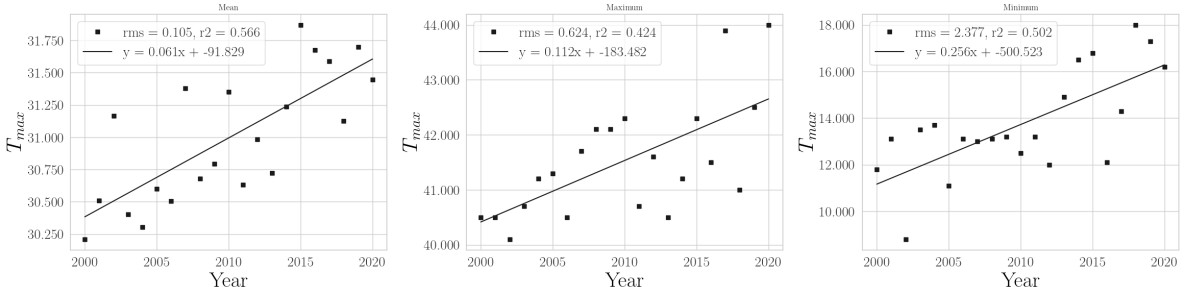

**Figure A1.** Annual mean, maximum, and minimum of daily maximum temperature in Midwest.

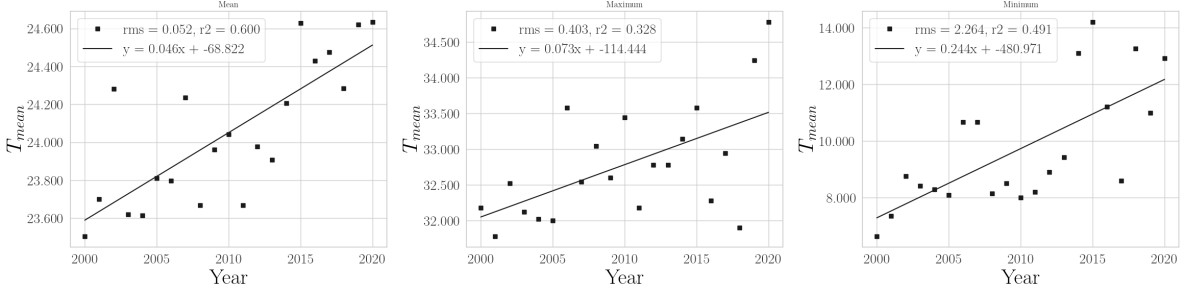

**Figure A2.** Annual mean, maximum, and minimum of daily mean temperature in Midwest.

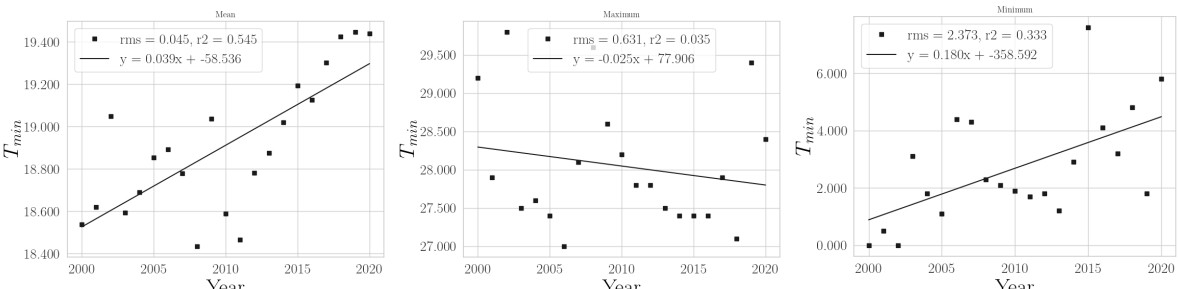

**Figure A3.** Annual mean, maximum, and minimum of daily minimum temperature in Midwest.

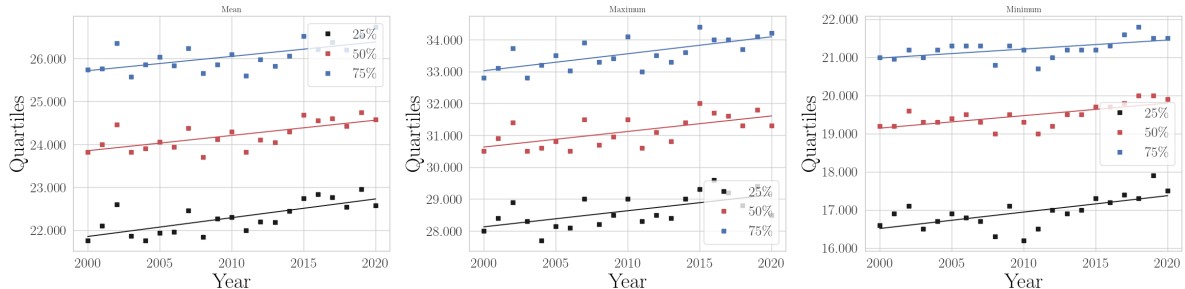

**Figure A4.** Annual quartiles for the mean, maximum, and minimum temperatures in Midwest.

Appendix A.1.2. Annual VPD and $ET_0$

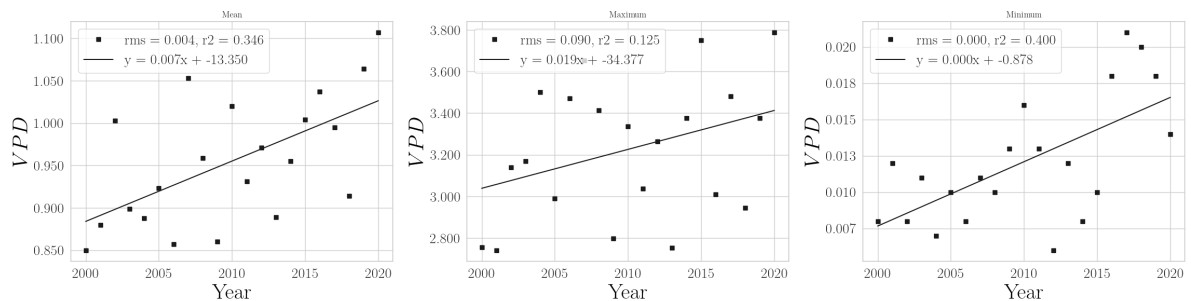

**Figure A5.** Annual mean, maximum, and minimum of VPD in Midwest.

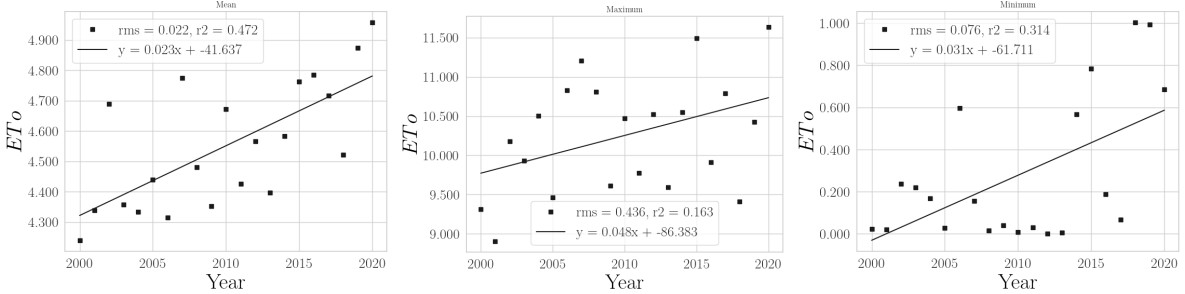

**Figure A6.** Annual mean, maximum, and minimum of $ET_0$ in Midwest.

*Appendix A.2. North*

Appendix A.2.1. Annual Temperatures

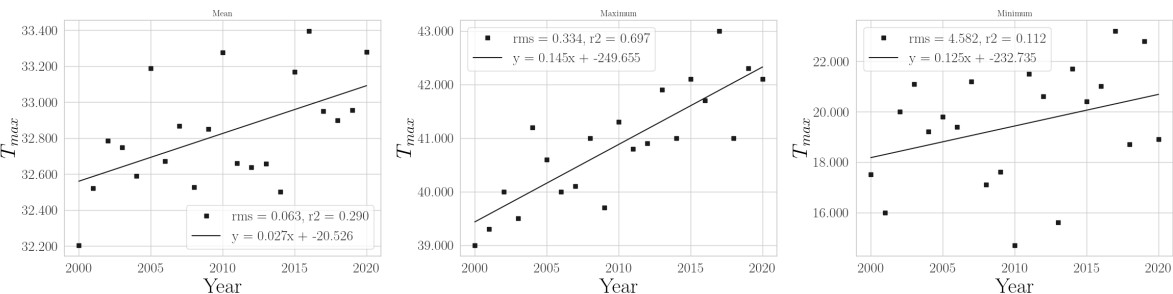

**Figure A7.** Annual mean, maximum, and minimum of daily maximum temperature in North.

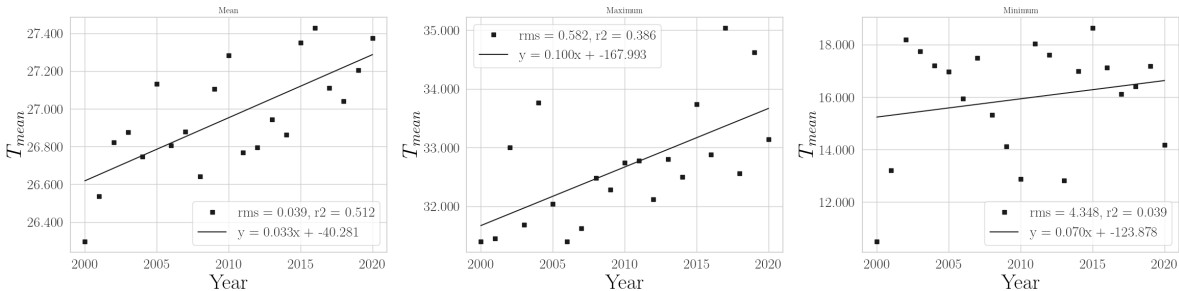

**Figure A8.** Annual mean, maximum, and minimum of daily mean temperature in North.

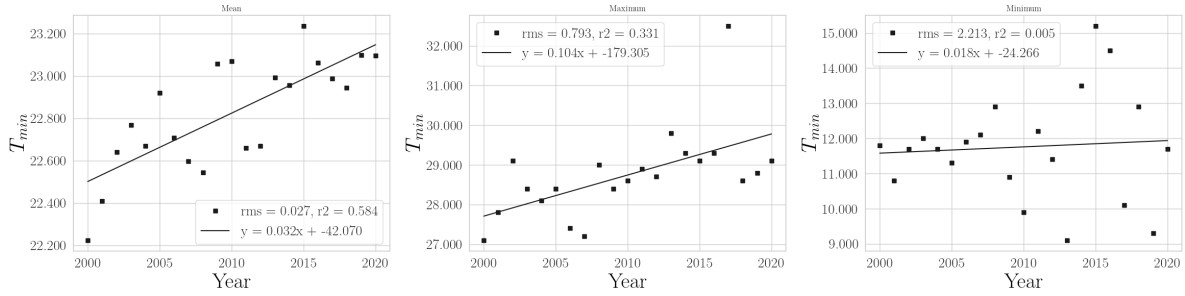

**Figure A9.** Annual mean, maximum, and minimum of daily minimum temperature in North.

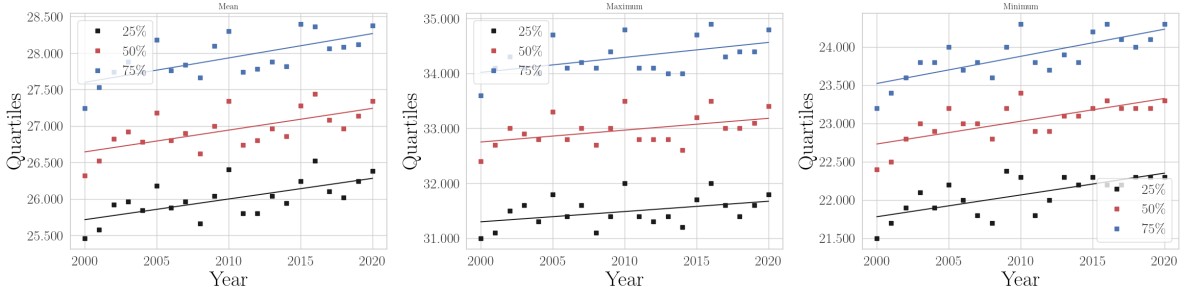

**Figure A10.** Annual quartiles for the mean, maximum, and minimum temperatures in North.

Appendix A.2.2. Annual VPD and $ET_0$

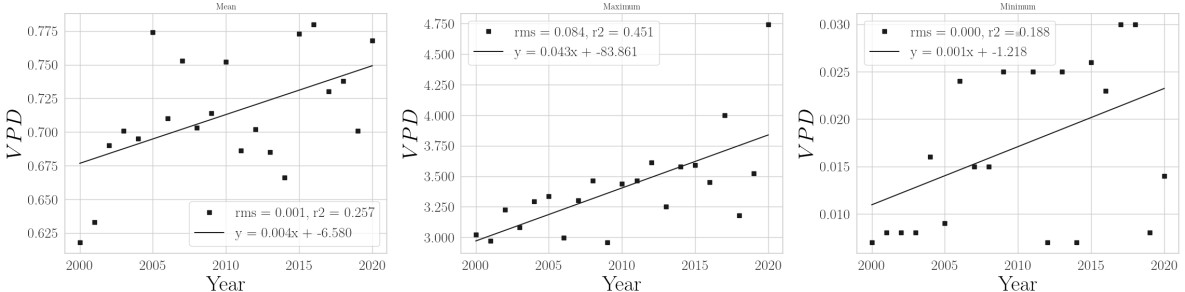

**Figure A11.** Annual mean, maximum, and minimum of VPD in North.

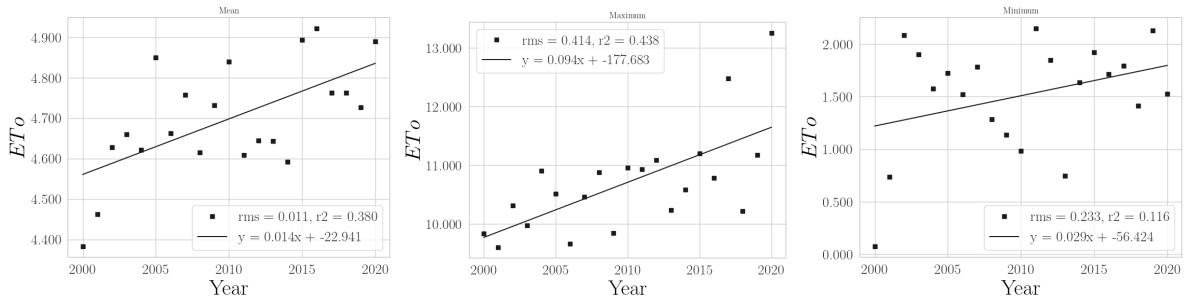

**Figure A12.** Annual mean, maximum, and minimum of $ET_0$ in North.

*Appendix A.3. Northeast*

Appendix A.3.1. Annual Temperatures

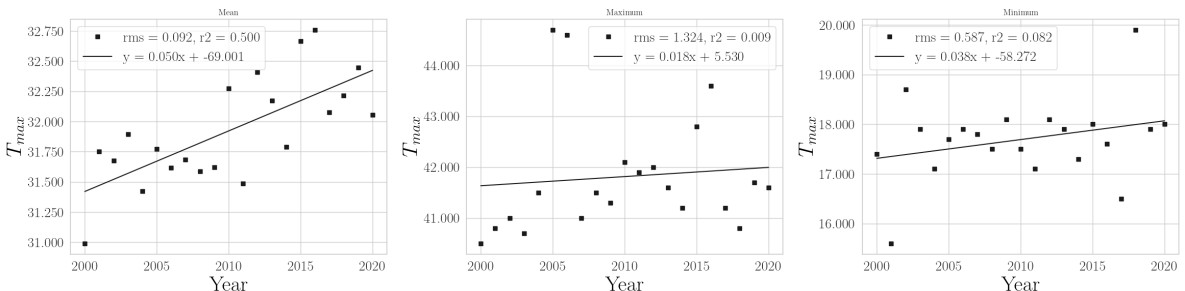

**Figure A13.** Annual mean, maximum, and minimum of daily maximum temperature in Northeast.

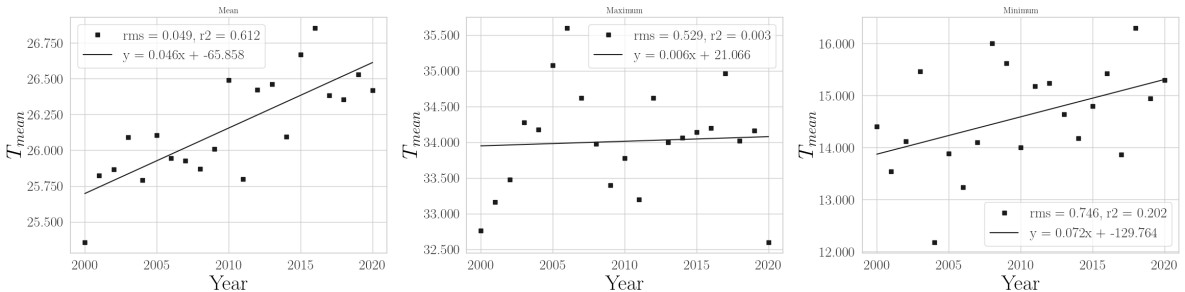

**Figure A14.** Annual mean, maximum, and minimum of daily mean temperature in Northeast.

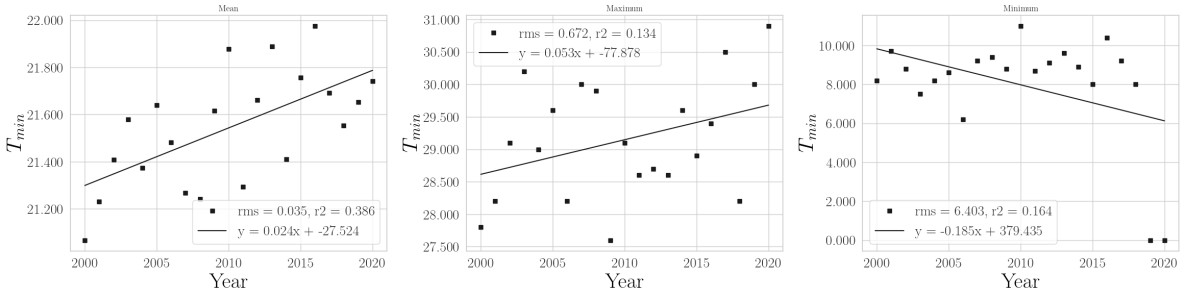

**Figure A15.** Annual mean, maximum, and minimum of daily minimum temperature in Northeast.

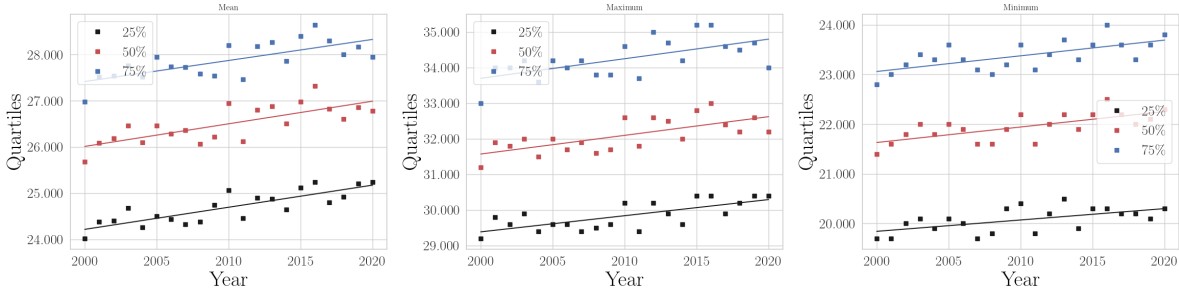

**Figure A16.** Annual quartiles for the mean, maximum, and minimum temperatures in Northeast.

Appendix A.3.2. Annual VPD and $ET_0$

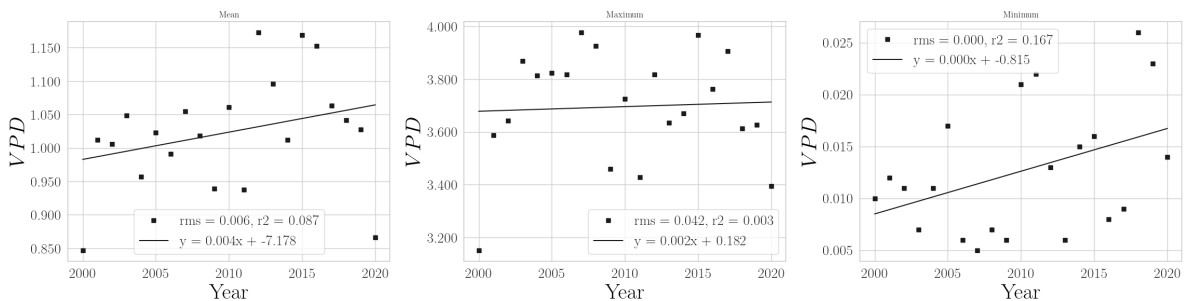

**Figure A17.** Annual mean, maximum, and minimum of VPD in Northeast.

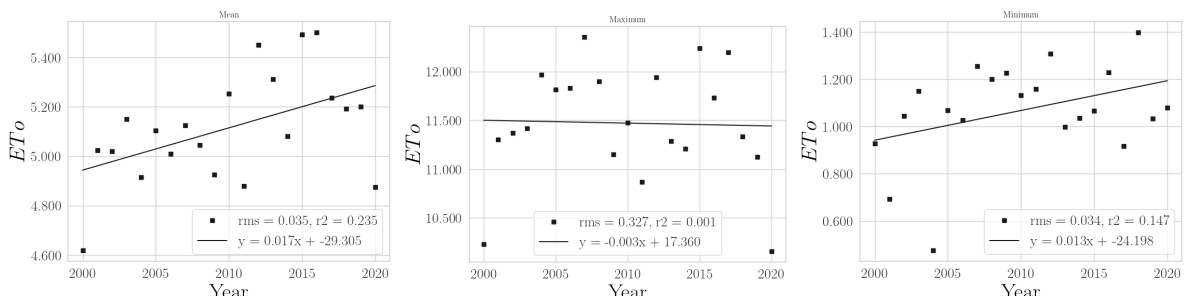

**Figure A18.** Annual mean, maximum, and minimum of $ET_0$ in Northeast.

*Appendix A.4. Southeast*

Appendix A.4.1. Annual Temperatures

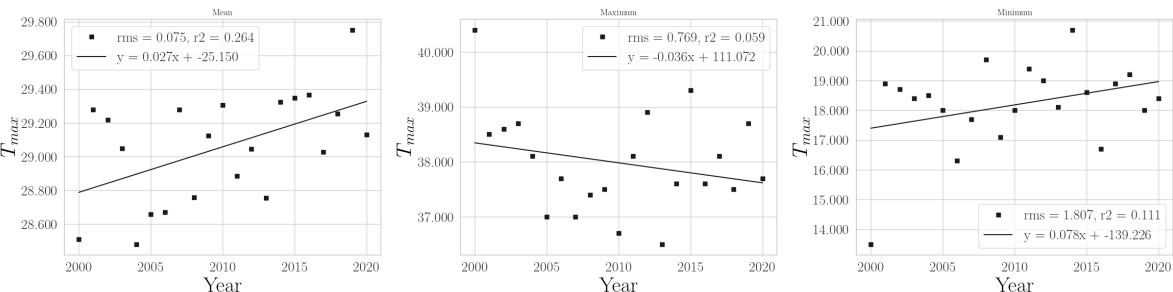

**Figure A19.** Annual mean, maximum, and minimum of daily maximum temperature in Southeast.

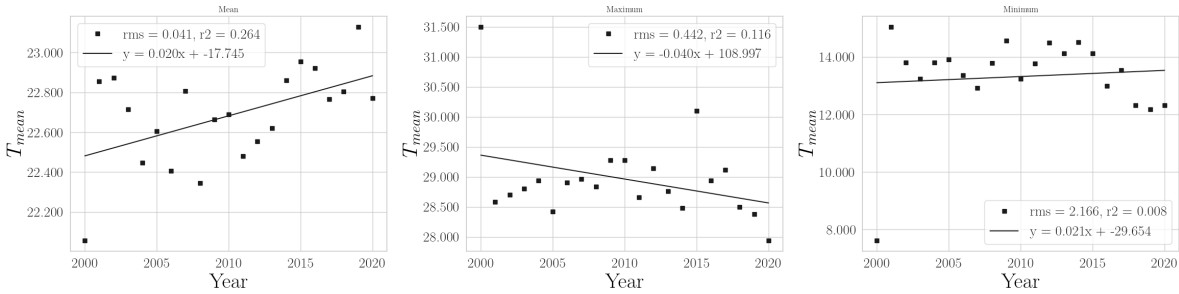

**Figure A20.** Annual mean, maximum, and minimum of daily mean temperature in Southeast.

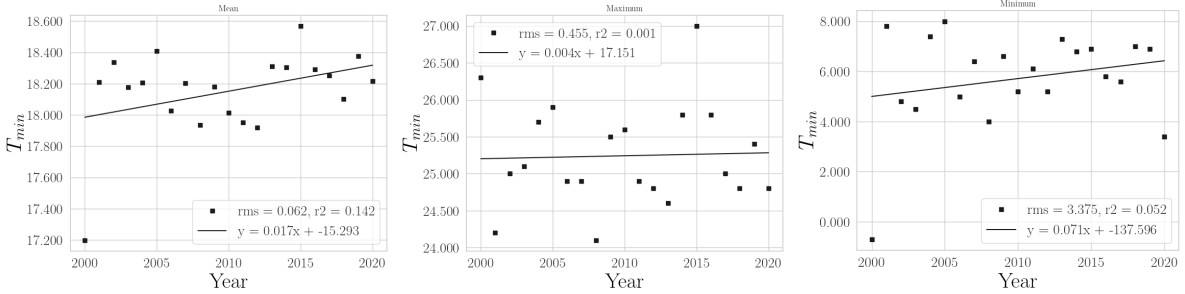

**Figure A21.** Annual mean, maximum, and minimum of daily minimum temperature in Southeast.

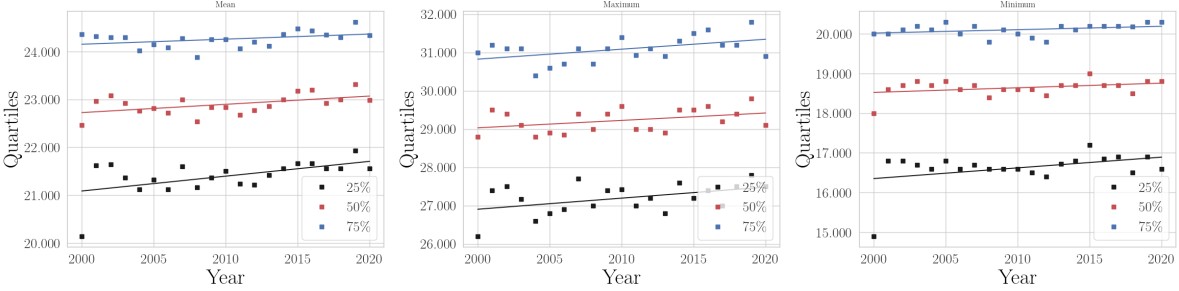

**Figure A22.** Annual quartiles for the mean, maximum, and minimum temperatures in Southeast.

Appendix A.4.2. Annual VPD and $ET_0$

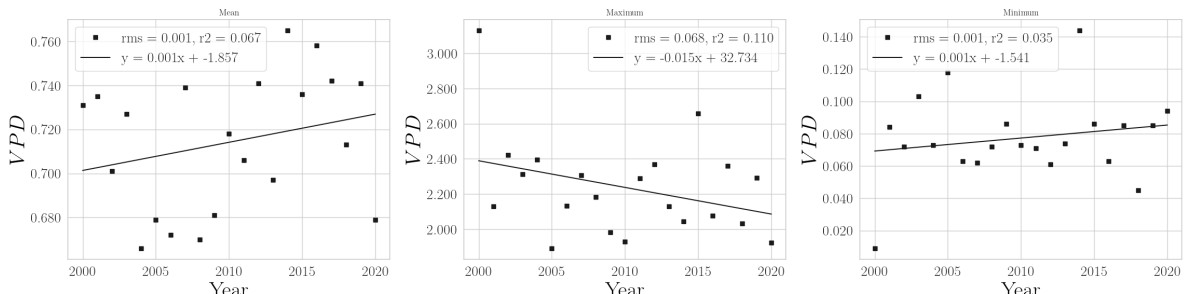

**Figure A23.** Annual mean, maximum, and minimum of VPD in Southeast.

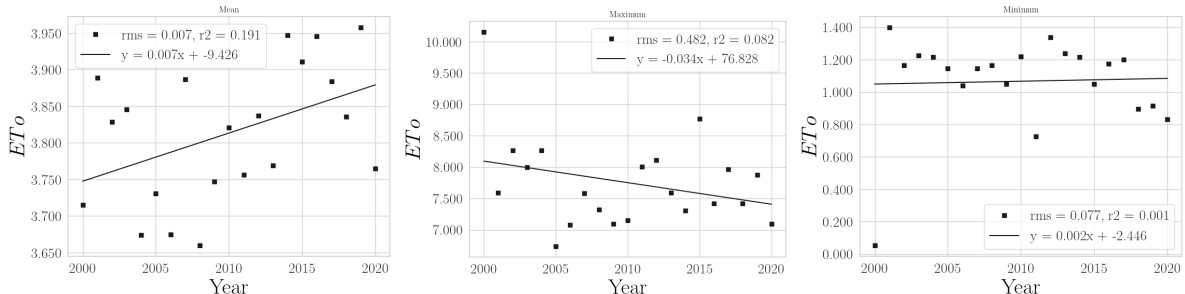

**Figure A24.** Annual mean, maximum, and minimum of $ET_0$ in Southeast.

*Appendix A.5. South*

Appendix A.5.1. Annual Temperatures

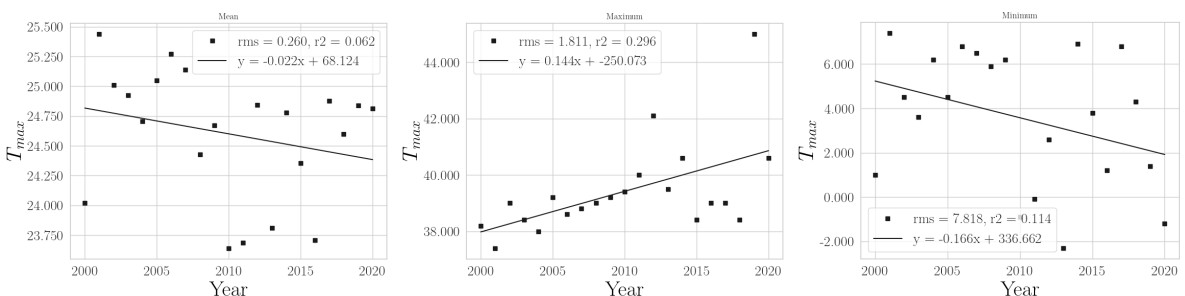

**Figure A25.** Annual mean, maximum, and minimum of daily maximum temperature in South.

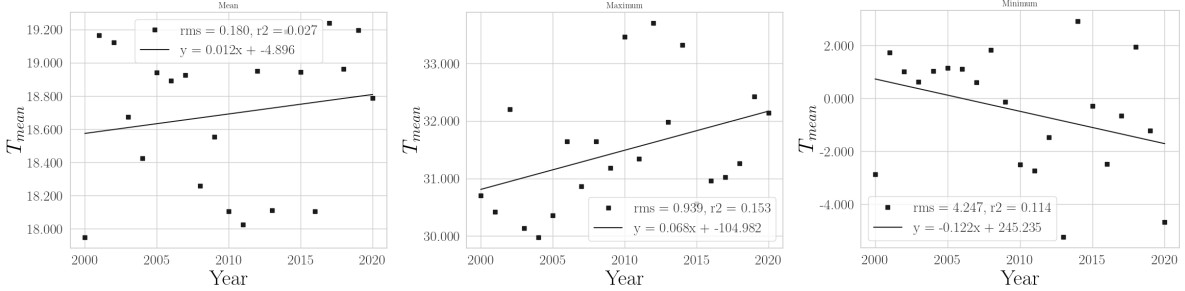

**Figure A26.** Annual mean, maximum, and minimum of daily mean temperature in South.

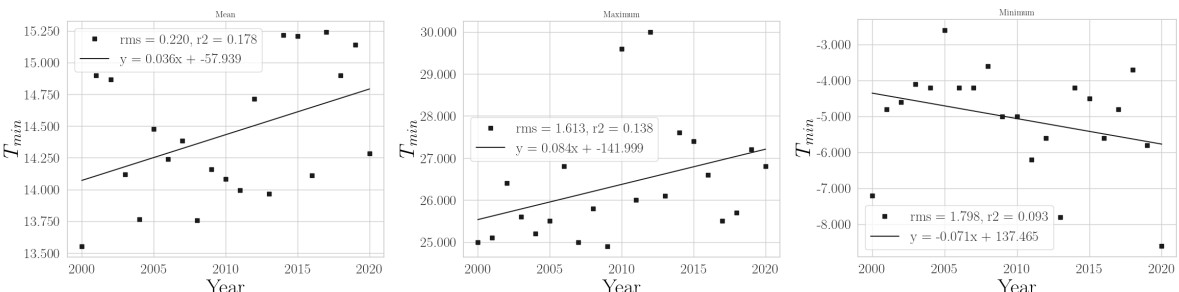

**Figure A27.** Annual mean, maximum, and minimum of daily minimum temperature in South.

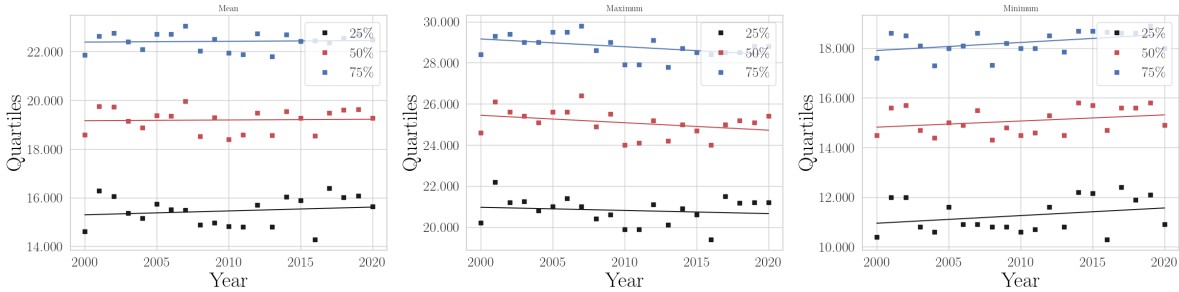

**Figure A28.** Annual quartiles for the mean, maximum and minimum temperatures in South.

Appendix A.5.2. Annual VPD and $ET_0$

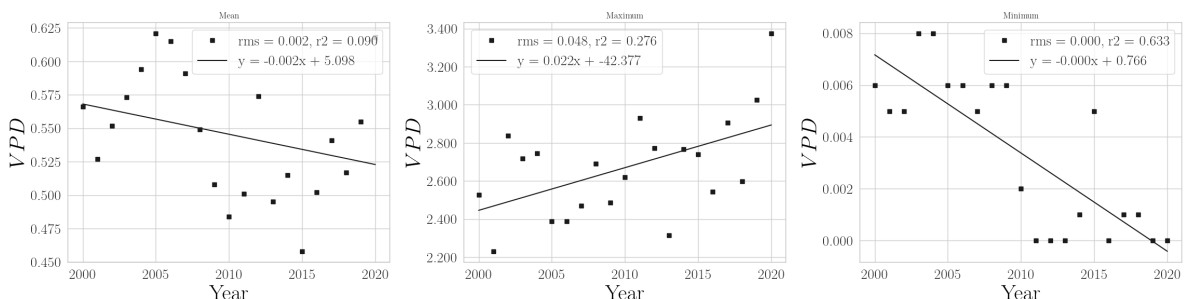

**Figure A29.** Annual mean, maximum, and minimum of VPD in South.

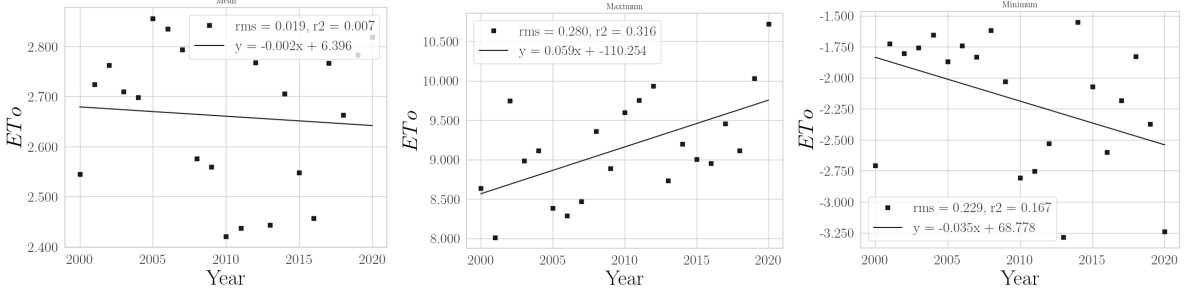

**Figure A30.** Annual mean, maximum, and minimum of $ET_0$ in South.

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
