# Peer review of "Trends and Patterns of Daily Maximum, Minimum and Mean Temperature in Brazil from 2000 to 2020"

_climate, doi:10.3390/cli11080168_

Round 1

Reviewer 1 Report

The article is devoted to temperature changes in different regions of Brazil. The article contains a lot of interesting information and therefore deserves publication. The disadvantage of the work is the difficulty of using this information, although the article contains more than a hundred drawings relating to the evolution of the temperature of individual regions. I consider it necessary to cite the statistical change in the average temperature of the whole of Brazil over 20 years, together with a statistical error, and add these figures to the annotation. In addition, the authors claim that one of the reasons for the temperature change is the destruction of forests. To show the degree of reliability of this statement, it is necessary to compare the average statistical temperature change over 20 years in the Mississippi region, where forests are being cut down, with this value for the whole of Brazil.

Author Response

Dear Reviewer 1

We thank you for your delicate and essential comments and suggestions for improving our manuscript. The revision was carried out, always based on the reviewer's comments, which were highly astute and guided us to improve the quality of our manuscript.

With the update, care has been taken to respond to each reviewer's questions, comments, and editorial requests. We point out that each reviewer's original comments are in black, and the author’s responses are in red.

We hope that the revisions in the manuscript and our accompanying responses meet the requirements for publication in CLIMATE. Please do not hesitate if you have further questions or concerns.

Thank you again for your consideration.

Thiago Rangel Rodrigues

on behalf of all the co-authors

Reviewer #1

Thank you for your feedback. All comments have been accepted and corrected according to the guidelines. Our corrections will be sufficient for the publication to be accepted.

Comments and Suggestions for Authors

The article is devoted to temperature changes in different regions of Brazil. The article contains much interesting information and therefore deserves publication. The disadvantage of the work is the difficulty of using this information. However, the article contains more than a hundred drawings relating to the evolution of the temperature of individual regions. I consider it necessary to cite the statistical change in the average temperature of the whole of Brazil over 20 years, together with a statistical error, and add these figures to the annotation. In addition, the authors claim that one of the reasons for the temperature change is the destruction of forests. To show the degree of reliability of this statement, it is necessary to compare the average statistical temperature change over 20 years in the Mississippi region, where forests are being cut down, with this value for the whole of Brazil.

Thank you for your relevant comment. The information about the statistical error has been introduced in the manuscript. See Lines 90-94.

And High discussion and comparison of the Mississippi region have been introduced in the manuscript. See Lines 262-282.

Reviewer 2 Report

Comments

Title

The title of the paper and its aims are different. The title mentions temperate ecosystems, but there is no analysis for such ecosystems in the aim.

Introduction

Line 25 “One of the dominant factors affecting weather patterns around the world is the so-called El Nino Southern Oscillation”. The phenomenon of El Nino Southern Oscillation is actively researched and its definition exists.

Material and methods

Line 75 Please add accessed date

Line 73-75 There is no information on the use of precipitation data. At the same time, precipitation is mentioned in lines 84-86

Line 79-80 It is missing decryption UR

Line 84-86 There is no reference to selected dry conditions criteria. If these criteria are chosen personally by the authors, then what determines them.

Figure 1 shows a map of Brazil divided into 5 regions. The Material and methods section does not contain information about such a separation. Why exactly these boundaries? Annual precipitation plots do not contain y-axis labels

According to the title and general objective, the paper should contain an analysis of drought index / drought rates. The section does not contain descriptions of drought index / drought rates.

Results

Line 103 It should be corrected “Figure XX”

Line 131 Decryption R2 should be done

Line 166-178 The results of the analysis of the linear trend of the considered values are presented. It is necessary to add an assessment of the significance of the obtained results. The Material and methods section does not contain information about the analysis of the linear trend

Line 171the Cerrado and agricultural activities predominate. atividades agrıcolas.”, there is a full stop after predominate.

Figure 3 shows the annual highest temperature. Under annual highest temperature is understood the absolute maximum value?

Figure 4 and Figure 8 caption should be corrected. In particular, figure 4a shows Tmax, 4b shows Tmean. The caption contains a description of Tmax. Figure 8 caption contain “DPV”, it should be VPD

The legends in Figure 2-8 need to be enlarged.

Discussion

Line 199 -200 The text needs to be corrected

Line 216 link “citecosta2020” should be corrected

Conclusion

The conclusions of the paper are very short. Conclusions about a significant increase of analyzed characteristics over 20 years are not supported by an assessment of the significance of the obtained results. Such an assessment is not given in the paper.

Author Response

Dear Reviewer 2

We thank you for your delicate and essential comments and suggestions for improving our manuscript. The revision was carried out, always based on the reviewer's comments, which were highly astute and guided us to improve the quality of our manuscript.

With the update, care has been taken to respond to each reviewer's questions, comments, and editorial requests. We point out that each reviewer's original comments are in black, and the author’s responses are in red.

We hope that the revisions in the manuscript and our accompanying responses meet the requirements for publication in CLIMATE. Please do not hesitate if you have further questions or concerns.

Thank you again for your consideration.

Thiago Rangel Rodrigues

on behalf of all the co-authors

Reviewer #2

Thank you for your feedback. All comments have been accepted and corrected according to the guidelines. Our corrections will be sufficient for the publication to be accepted.

Comments

Title

The title of the paper and its aims are different. The title mentions temperate ecosystems, but there is no analysis for such ecosystems in the aim.

First, thank you for your comments. The title of the article has been rewritten.

Introduction

Line 25 “One of the dominant factors affecting weather patterns around the world is the so-called El Nino Southern Oscillation”. The phenomenon of El Nino Southern Oscillation is actively researched and its definition exists.

Material and methods

Line 75 Please add accessed date

It has been Added. Lines 78-79

Line 73-75 There is no information on the use of precipitation data. At the same time, precipitation is mentioned in lines 84-86

It has been Added. Lines 76-78

Line 79-80 It is missing decryption UR

Decryption UR is in lines 84-85

Line 84-86 There is no reference to selected dry conditions criteria. If these criteria are chosen personally by the authors, then what determines them.

It has been added in lines 90-92

Figure 1 shows a map of Brazil divided into five regions. The Material and methods section does not contain information about such a separation. Why exactly these boundaries? Annual precipitation plots do not contain y-axis labels

It has been added in lines 74-76 and in Figure 1

According to the title and general objective, the paper should contain an analysis of drought index / drought rates. The section does not contain descriptions of drought index / drought rates.

The title of the article has been rewritten

Results

Line 103 It should be corrected “Figure XX”

It has been adjusted.

Line 131 Decryption R2 should be done

The information about the statistical error, has been introduced in the manuscript. Lines 90-94.

Line 166-178 The results of the analysis of the linear trend of the considered values are presented. It is necessary to add an assessment of the significance of the obtained results. The Material and methods section does not contain information about the analysis of the linear trend

The information about the statistical error, has been introduced in the manuscript. Lines 90-94.

Line 171” the Cerrado and agricultural activities predominate. atividades agrıcolas.”, there is a full stop after predominate.

It has been  adjusted .

Figure 3 shows the annual highest temperature. Under annual highest temperature is understood the absolute maximum value?

It has been  adjusted in line 197

Figure 4 and Figure 8 caption should be corrected. In particular, figure 4a shows Tmax, 4b shows Tmean. The caption contains a description of Tmax. Figure 8 caption contain “DPV”, it should be VPD

It has been  adjusted

The legends in Figure 2-8 need to be enlarged.

It has been  adjusted

Discussion

Line 199 -200 The text needs to be corrected

It has been  adjusted

Line 216 link “citecosta2020” should be corrected

It has been  adjusted

Conclusion

The conclusions of the paper are very short. Conclusions about a significant increase of analyzed characteristics over 20 years are not supported by an assessment of the significance of the obtained results. Such an assessment is not given in the paper.

It has been introduced in the manuscript. Lines 291-306.

Round 2

Reviewer 2 Report

article has improved a lot. in further studies, when assessing the significance of the results obtained, I recommend using well-known criteria and methods. for example, use the least squares method to evaluate the significance of linear trends. the coefficient of determination does not fully determine the significance of the trend.

Author Response

Thank you for your feedback. All comments have been accepted and corrected according to the guidelines. Our corrections will be sufficient for the publication to be accepted.

Comments and Suggestions for Authors

Comments

Comments and Suggestions for Authors

Article has improved a lot. in further studies, when assessing the significance of the results obtained, I recommend using well-known criteria and methods. for example, use the least squares method to evaluate the significance of linear trends. the coefficient of determination does not fully determine the significance of the trend.

First, thank you for your comments. A statistical regression analysis has been included in the Manuscript in Table 3, see line 215.
